# A Study on the Formation Reactions and Conversion Mechanisms of HONO and HNO₃ in the Atmosphere of Daejeon, Korea

Kyoungchan Kim [1], Chunsang Lee [1], Dayeong Choi [1], Sangwoo Han [2], Jiwon Eom [3] and Jinseok Han [1,*]

[1] Department of Environmental Engineering, Anyang University, Anyang 14028, Republic of Korea; rudcksdlqk@naver.com (K.K.)
[2] E2M3 Inc., Anyang 14059, Republic of Korea
[3] Department of Chemistry, Yonsei University, Seoul 03722, Republic of Korea
* Correspondence: nierhan@hanmail.net; Tel.: +82-31-463-1292

**Abstract:** Nitrogen oxides ($NO_X$) in the atmosphere cause oxidation reactions with photochemical radicals and volatile organic compounds, leading to the accumulation of ozone ($O_3$). $NO_X$ constitutes a significant portion of the $NO_y$ composition, with nitrous acid (HONO) and nitric acid ($HNO_3$) following. HONO plays a crucial role in the reaction cycle of $NO_X$ and hydrogen oxides. The majority of $HNO_3$ reduction mechanisms result from aerosolization through heterogeneous reactions, having adverse effects on humans and plants by increasing secondary aerosol concentrations in the atmosphere. The investigation of the formation and conversion mechanisms of HONO and $HNO_3$ is important; however, research in this area is currently lacking. In this study, we observed HONO, $HNO_3$, and their precursor gases were observed in the atmosphere using parallel-plate diffusion scrubber-ion chromatography. A 0-D box model simulated the compositional distribution of $NO_y$ in the atmosphere. The formation reactions and conversion mechanisms of HONO and $HNO_3$ were quantified using reaction equations and reaction coefficients. Among the various mechanisms, dominant mechanisms were identified, suggesting their importance. According to the calculation results, the produce of HONO was predominantly attributed to heterogeneous reactions, excluding an unknown source. The sink processes were mainly governed by photolysis during daytime and reactions with OH radicals during nighttime. $HNO_3$ showed dominance in its production from $N_2O_5$, and in its conversion mechanisms primarily involving aerosolization and deposition.

**Keywords:** HONO; $HNO_3$; formation and conversion mechanisms; F0AM; $NO_y$; ion chromatography; PTR-ToF-MS

## 1. Introduction

Nitrogen oxides ($NO_X$), the sum of nitric oxide (NO) and nitrogen dioxide ($NO_2$), cause the oxidation of photochemical radicals and volatile organic compounds (VOCs) in the troposphere [1,2]. $NO_y$, is a collective abbreviation for atmospheric nitrogen oxides, consisting of $NO_Z$ and $NO_X$. $NO_Z$, nitrogen oxides excluding $NO_X$, act as a reservoir for $NO_X$ [3]. $NO_2$ and NO account for the largest composition of $NO_y$, followed by nitrous acid (HONO), nitric acid ($HNO_3$), and other $NO_Z$ species [4].

HONO is an important atmospheric compound because of its contribution to the reaction cycle of $NO_X$ and hydrogen oxide radicals ($HO_x$) [4,5]. Photolysis of HONO in the near-ultraviolet spectral range (<320 nm, >400 nm) produces OH radicals and NO, regardless of the amount of ozone ($O_3$) photolysis [6,7]. During the day, photolysis is a major reduction mechanism for HONO [1,8,9]. The OH radicals generated by the photolysis of HONO play a role in triggering the accumulation of $O_3$ in the atmosphere, which substantially impacts the occurrence of photochemical smog in urban areas. At night, HONO is primarily known to be produced through the gas–liquid heterogeneous reaction of

$NO_2$. To investigate this reaction, studies have been conducted to determine the relationship between nighttime $NO_2$ and relative humidity [10–12]. Despite the importance of HONO in atmospheric chemistry, detailed research on the reaction mechanisms of HONO in the atmosphere is lacking. This is because the yet-to-be-identified reactions during discharge and the homogeneous and heterogeneous reaction processes have complex effects on HONO concentrations [7,13,14].

Atmospheric $HNO_3$ is formed through various pathways, primarily through the reaction of $NO_2$ and OH during the day, and $N_2O_5$ and $H_2O$ at night, being the most important production reactions [15–17]. In the reduction pathway, aerosolization via a heterogeneous reaction plays a significant role [16,17]. Heterogeneous reactions are vital in both the stratosphere and the troposphere, contributing to the increase in secondary particulate matter concentration in the atmosphere [18,19]. $HNO_3$ and sulfuric acid gases ($H_2SO_4$) undergo a heterogeneous reaction with ammonia gas ($NH_3$), converting them into secondary ultrafine particles, namely ammonium nitrate ($NH_4NO_3$) and ammonium sulfate (($NH_4$)$_2SO_4$).

The aerosolization mechanism of $HNO_3$ predominantly contributes to the reduction in the $HNO_3$ concentration, along with other reduction mechanisms such as drying and wet deposition [19,20]. Studies have been conducted to investigate the conversion mechanisms of $HNO_3$ into particulate matter and precursors [17,18,20–22]. However, further research is necessary to comprehensively explain and quantify the process of converting gaseous $HNO_3$ concentration into the particulate phase.

Gil et al. (2020) used the parallel plate diffusion scrubber-ion chromatograph (PPDS-IC) system to measure the $HNO_3$ concentrations in the atmosphere [14]. In addition, the OH generation rate of $HNO_3$ was calculated using the Framework for 0-D Atmospheric Modeling (F0AM) model, a 0-D box model, by comparing photochemical pollution case days with non-case days according to $O_3$ concentration. The research concluded that the accumulation of $O_3$ proceeded by producing OH radicals via the photolysis of $HNO_3$ in the early morning. However, this work focused on photolysis of $HNO_3$, and further expanded studies on the mechanisms of production or formation of $HNO_3$ are needed. Studies that can estimate the mechanism of formation and conversion reaction of $HNO_3$ in detail are needed. Chou et al. (2009) evaluated the effect of $O_3$ production on $NO_y$, which has a major effect on the chemical reaction of $O_3$ [3]. In this study, $NO_y$ was divided into $NO_X$ and $NO_Z$, resulting in the $O_3$ production efficiency of $NO_X$, and the $NO_Z$ and $O_3$ concentrations were positively correlated. Throughout the investigation, they assessed alterations in the composition ratio of $NO_Z$, NO, and $NO_2$. However, there is a gap in the existing literature as no previous studies have delved into the composition ratio and distribution characteristics of individual species, namely HONO and $HNO_3$, $N_2O_5$, $NO_3$, and peroxyacetyl nitrate (PAN), which collectively constitute $NO_Z$. Hou et al. (2016) and Liu et al. (2021) measured the HONO concentration in an urban area during the summer months, and the concentration was evaluated along with various variable factors [11,12]. In addition, the formation and uptake reactions of HONO were analyzed in detail to estimate several emission sources and conversion reactions, including unknown sources of HONO. As such, there are many studies that have evaluated and estimated the different reaction mechanisms of HONO; however, studies on $HNO_3$ are lacking. Watson et al. (1994) used the SEQUILIB model, a thermodynamic equilibrium model of the secondary aerosol, and simulated and compared the reduction in aerosol precursors in the winter months in Arizona with measurement results [17]. In addition, an equilibrium equivalence concentration curve of nitrates in the particulate component that varied with the humidity levels was presented. However, the application of these models was limited to Arizona, and changes in equilibrium with temperature levels were not considered. Therefore, it is necessary to develop a mechanism that can explain the aerosol conversion process of $HNO_3$ more closely, and if it can be applied to the gas–particle equilibrium model, it will be a more powerful research tool.

In this study, the concentrations of $NO_X$ and intermediate products of particulate matter (HONO and $HNO_3$) were measured. The production and conversion mechanisms of the pollutants were also analyzed. Seasonal differences were examined by comparing concentrations in winter and summer. The characteristics of the relationship between HONO, $NO_2$, and relative humidity, essential in the initial photochemical reactions of diurnal patterns, were investigated. Furthermore, to focus on the effect of changes in the concentrations of HONO and OH radicals on $O_3$ in the atmosphere during summer, when photochemical reactions occur actively, the F0AM model was used for a quantitative evaluation of the production and reduction reactions of HONO and $HNO_3$.

## 2. Materials and Methods

### 2.1. Sampling Site and Duration

Field measurements were performed at the Central Intensive Air-monitoring Site in Jungangro 12, Junggu, Daejeon, Republic of Korea (36.322° N, 127.414° E). The site is located near an expressway that is severely affected by vehicle traffic and biomass-burning activities, including the incineration of agricultural waste, frequently occurring in the surrounding area (Figure 1). Measurements were carried out for 23 days from 7 to 29 January 2021, for the winter season and 28 days from 18 May to 16 June 2021, for the summer season.

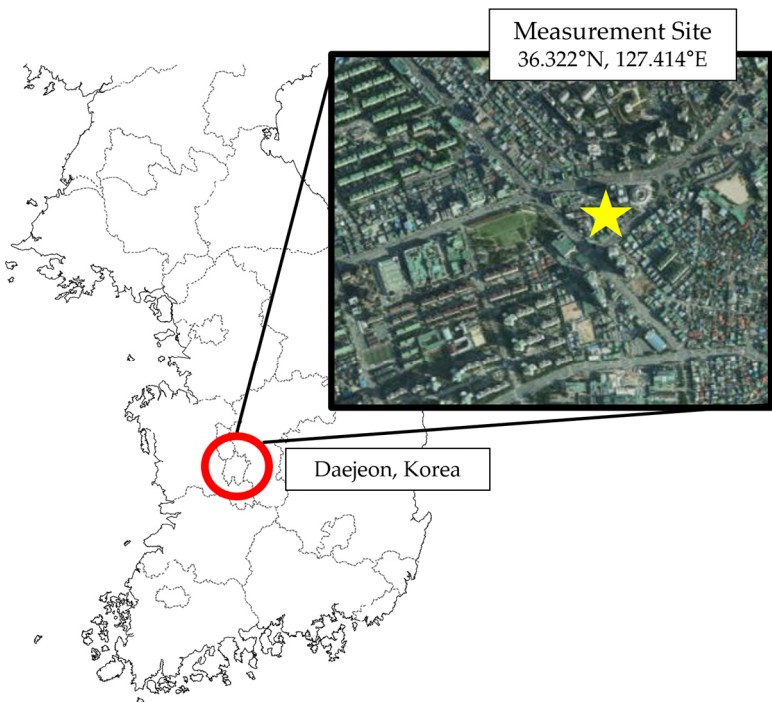

**Figure 1.** Location of the Central Intensive Air-monitoring Site in Daejeon, Republic of Korea. The yellow star indicates the measurement position.

### 2.2. PPDS-IC System

In this study, ambient air was analyzed using the PPDS-IC method, which is schematically shown in Figure 2. A membrane positioned between the liquid and air channels, preventing the passage of particles from the air into the liquid channel. Instead, gas molecules soluble in water pass through the membrane, dissolve in the distillation water flowing through the liquid channel, and the resulting sample is then transported to the IC for measurement [23].

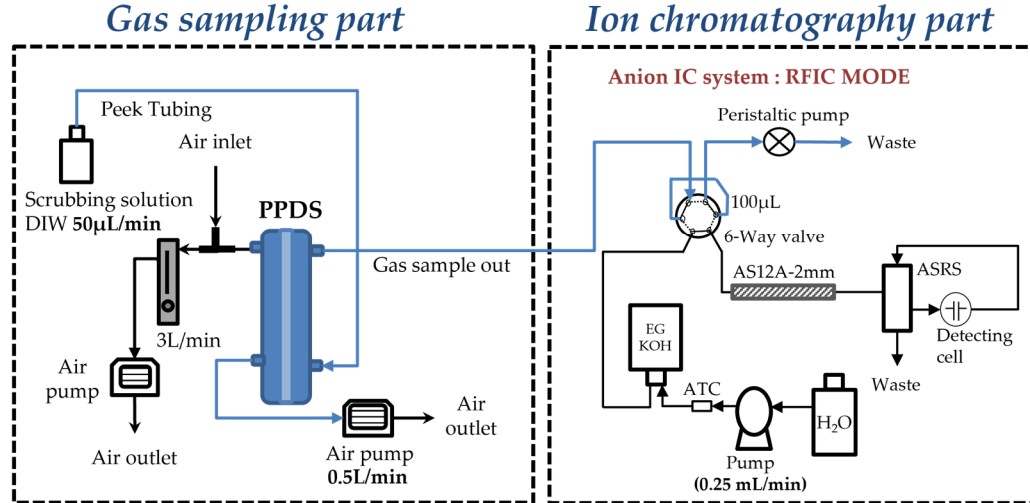

**Figure 2.** Schematics of the PPDS-IC systems.

The absorbent flowed through the channel at a constant flow rate of 50 μL/min, and the absorbent solution was injected every hour using the sample autoinjector of IC systems. Cations and anions were eluted with 10 mM methane sulfonic acid (MSA; J.T. Baker, Phillipsburg, NJ, USA) and 40 mM KOH (J.T. Baker, Phillipsburg, NJ, USA), respectively. An IC system analyzer (CD20, DIONEX, Sunnyvale, CA, USA) operated using reagent-free ion chromatography with an eluent generation mode. To eliminate bubbles that could arise due to temperature fluctuations in deionized water during measurement, continuous purging was performed using helium gas (JC GAS, Gyeonggi, Republic of Korea). The PPDS-IC measurement method details are outlined by Kim et al. (2021) [24], and the IC conditions for the measurements are provided in Table 1.

**Table 1.** Analysis conditions of IC.

| IC | Cation | Anion |
|---|---|---|
| Analytical System | Dionex CD20 | Dionex CD20 |
| Analytical Column | IonPac CS15 (2 × 250 mm, Dionex) | IonPac AS12A (2 × 250 mm, Dionex) |
| Eluent | MSA 10 mM (in RFIC mode) | KOH 40 mM (in RFIC mode) |
| Eluent Flow Rate | 0.25 mL/min | 0.25 mL/min |
| Cell Temperature | 35 °C | 35 °C |
| Injection Volume | 500 μL | 100 μL |
| Suppressor | CERS (in Recycle mode) | AERS (in Recycle mode) |
| Suppressor Current | 11 mA | 25 mA |
| Background Conductivity | 274 nS/cm | 334 nS/cm |
| Pressure | 1210 psi | 1762 psi |

### 2.3. Box Model (F0AM)

Whereas direct measurements of OH and $HO_2$ radicals and $NO_3$, $N_2O_5$, and PAN were not performed in this study, the F0AM was used to calculate the mixing ratios of these precursor species. The F0AM is an open platform for simulating atmospheric chemistry [25]. The F0AM is mainly used to quantify the production and loss of reactants in chemical reactions involving numerous chemical and physical processes in the atmosphere [26]. The 0-D box model provides simplicity and ease of use but has limitations. It excludes the horizontal and vertical transport of atmospheric matter, and its reliability is not absolute. The model does not offer a comprehensive quantitative evaluation of physical conversion processes, such as aerosol formation or deposition. The F0AM is written in MATLAB and

has the option of selecting a chemical reaction based on several mechanisms, including the Master of Chemical Mechanism (MCM), the Carbon Bond mechanism, and the Regional Atmospheric Chemistry Mechanism. This study employed MCM v3.3.1, utilizing a 1 h average of the observed concentration and meteorological dataset [27,28]. Detailed chemical and photochemical reaction data for this mechanism can be accessed on the MCM website (https://mcm.york.ac.uk/MCM/ (accessed on 22 February 2024)). Only the summer data of the precursors were simulated using the F0AM because photochemical reactions occur more actively in summer.

### 2.4. PTR-ToF-MS

The key parameters in atmospheric photochemical reactions with $NO_X$ are the VOCs. A proton transfer reaction time-of-flight mass spectrometer (PTR-ToF-MS 1000) (IONICON, Innsbruck, Austria) was used for VOC observations. The biggest advantage of the PTR-ToF-MS is that atmospheric VOCs can be analyzed in real time without pretreatment. The detailed operation method used in this study has been described in a previous study [29–31]. In this study, calibration was conducted using standard gas before use and was used for measurement after a reliability test was performed [30].

### 2.5. Other Data

Meteorological data (temperature, relative humidity, pressure) were obtained from the Daejeon Meteorological Observatory automatic observation system, which can be accessed from the Korea Meteorological Administration website. (https://data.kma.go.kr/cmmn/main.do (accessed on 22 February 2024)). In addition, $SO_2$, NO, $NO_2$, CO, and $O_3$ were not measured directly; however, observational data from the same period provided by an air quality monitoring station (situated approximately 2.3 km away from the measurement site) operated by the National Institute of Environment and Research were used.

## 3. Results and Discussion

### 3.1. Data Overview

The mixing ratios of HONO in winter and summer were $2.59 \pm 1.91$ ppbv (n = 490) and $1.8 \pm 0.76$ ppbv (n = 687), respectively. Those of $HNO_3$ in winter and summer were $0.72 \pm 0.61$ ppbv (n = 375) and $0.1 \pm 0.03$ ppbv (n = 686), respectively. The observed mixing ratio of $O_3$ at the air quality monitoring station was $18 \pm 16$ ppbv (n = 525), and that of $NO_X$ was $56 \pm 55$ ppbv (n = 522). The $O_3$ and $NO_X$ mixing ratios in summer were $44 \pm 18.3$ ppbv (n = 687) and $15.9 \pm 8.5$ ppbv (n = 687), respectively.

Table 2 summarizes the mixing ratio distribution of gaseous matter observed during the measurement period. The average mixing ratio of HONO in winter was approximately 1.43 times higher than that in summer, and that of $HNO_3$ in summer was approximately 7.2 times higher than that in winter. The levels of mixing ratio of HONO in this study were higher than those observed by Chang et al. (2008) in Gwangju (0.5 ppbv in spring) and those measured by Gil et al. (2020) in Seoul (0.28 ppbv in summer) [14,32]. Ahn et al. (2013) measured $HNO_3$ in Seoul as 0.83 ppbv, which is higher than that in this study at Daejeon in winter (0.72 ppbv) but lower than that in summer (0.1 ppbv) [33]. The difference between the mentioned air pollutants may be due to the seasonal variability of anthropogenic load and the difference in the sources of pollutants themselves.

Figures 3 and 4 represent the time-series distribution for the entire measurement period in winter and summer, respectively. There was an absence of data because of the inspection of the measuring instrument and data below the detection limit. In winter, high HONO and $O_3$ concentrations were observed between 13 and 14 January. From 15 to 16 January, $PM_{2.5}$, HONO, $HNO_3$, and $O_3$ exhibited high concentrations. In summer, the $PM_{2.5}$ concentration was severe from 24 to 25 May; however, the measured mixing ratios of HONO and $HNO_3$ were relatively low. Diurnal distributions of HONO and $O_3$ were clearly observed. The average $NO/NO_2$ ratio in winter (1.4) was approximately seven times higher than that in summer (0.2).

**Table 2.** Summary of measurement data.

|  |  | Average | Max | Min | STD | n |
|---|---|---|---|---|---|---|
| HONO | winter | 2.59 * | 11.8 | 0.29 | 1.91 | 490 |
|  | summer | 1.8 | 5.6 | 0.3 | 0.76 | 687 |
| HNO$_3$ | winter | 0.72 | 3.1 | 0.03 | 0.61 | 375 |
|  | summer | 0.1 | 0.2 | 0.07 | 0.03 | 686 |
| PM$_{2.5}$ | winter | 20 | 86 | 1 | 12 | 445 |
|  | summer | 19.3 | 62 | 1 | 10.2 | 657 |
| O$_3$ | winter | 18 | 53 | 1 | 16 | 525 |
|  | summer | 44 | 99 | 5 | 18.3 | 687 |
| NO$_2$ | winter | 23 | 64 | 4 | 12 | 522 |
|  | summer | 13 | 38 | 5 | 5.5 | 687 |
| NO | winter | 33 | 288 | 0.1 | 43 | 483 |
|  | summer | 2.9 | 24.9 | 0.003 | 3.0 | 447 |
| CO | winter | 570 | 1700 | 100 | 250 | 525 |
|  | summer | 523 | 1000 | 300 | 94 | 687 |
| SO$_2$ | winter | 3 | 7 | 1 | 1.1 | 522 |
|  | summer | 2.8 | 10 | 0.1 | 10.1 | 687 |
| Temp. | winter | −0.12 | 14 | −17.3 | 7.4 | 525 |
|  | summer | 21.1 | 32.7 | 11.9 | 4.7 | 687 |
| R.H. | winter | 67.1 | 97 | 18 | 19 | 525 |
|  | summer | 74.3 | 97 | 23 | 18.8 | 687 |

* The unit for PM$_{2.5}$ is µg/m$^3$, that for Temp. (Temperature) is °C, that for relative humidity is %, and that for others is ppbv.

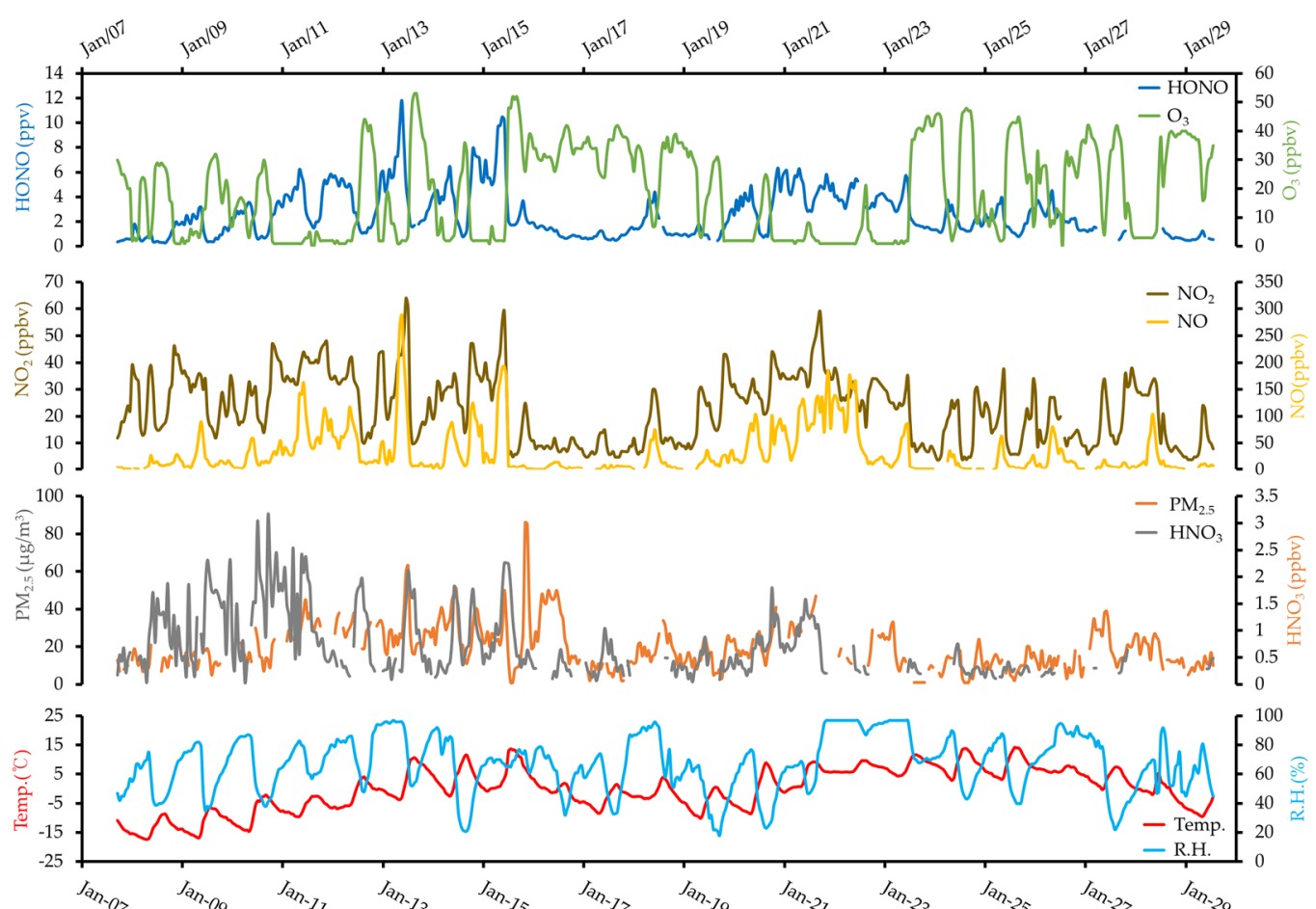

**Figure 3.** Time−series distribution for measurements in winter.

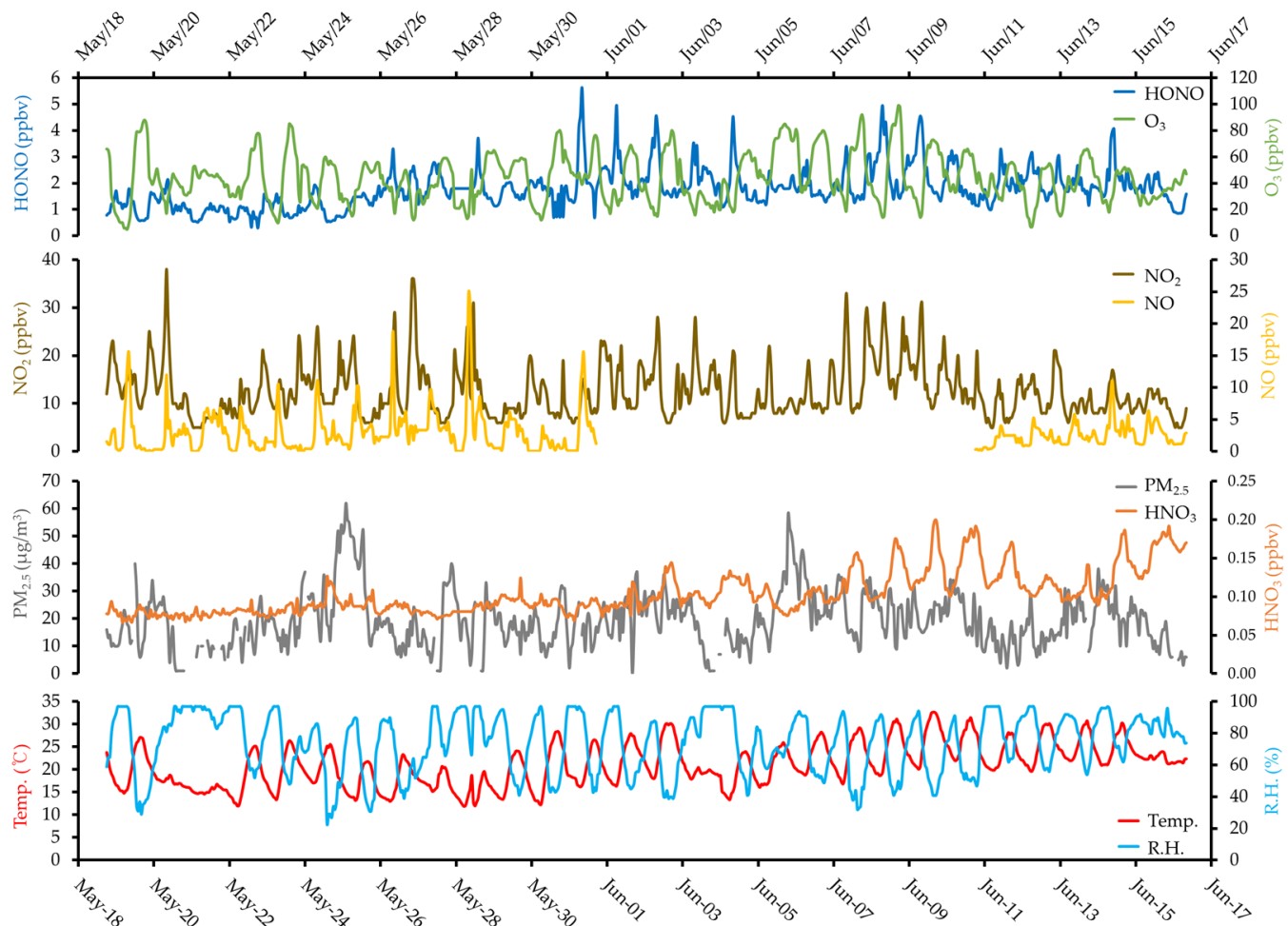

**Figure 4.** Time−series distribution for measurements in summer.

The heightened concentrations of air pollutants in Daejeon during winter, compared to summer, may be attributed, in part, to prevailing westerly winds facilitating the transport of emissions from major industrial facilities and power plants located to the northwest. This geographical arrangement, coupled with seasonal weather patterns, likely contributes to an increased impact on air quality during winter in the region.

Figure 5 illustrates the diurnal variations in the average concentrations of the measured pollutants during winter and summer. The HONO concentration in both seasons sharply decreased from 7 to 8 a.m. because of photochemical reactions. At sunrise, $O_3$ concentrations exhibited a rapid increase. NO and $NO_2$ showed high concentrations during the morning hours, indicating the influence of nearby vehicular emission sources. $HNO_3$ increased in the morning and gradually decreased after noon during winter, whereas during summer, it increased until the evening and decreased after sunset.

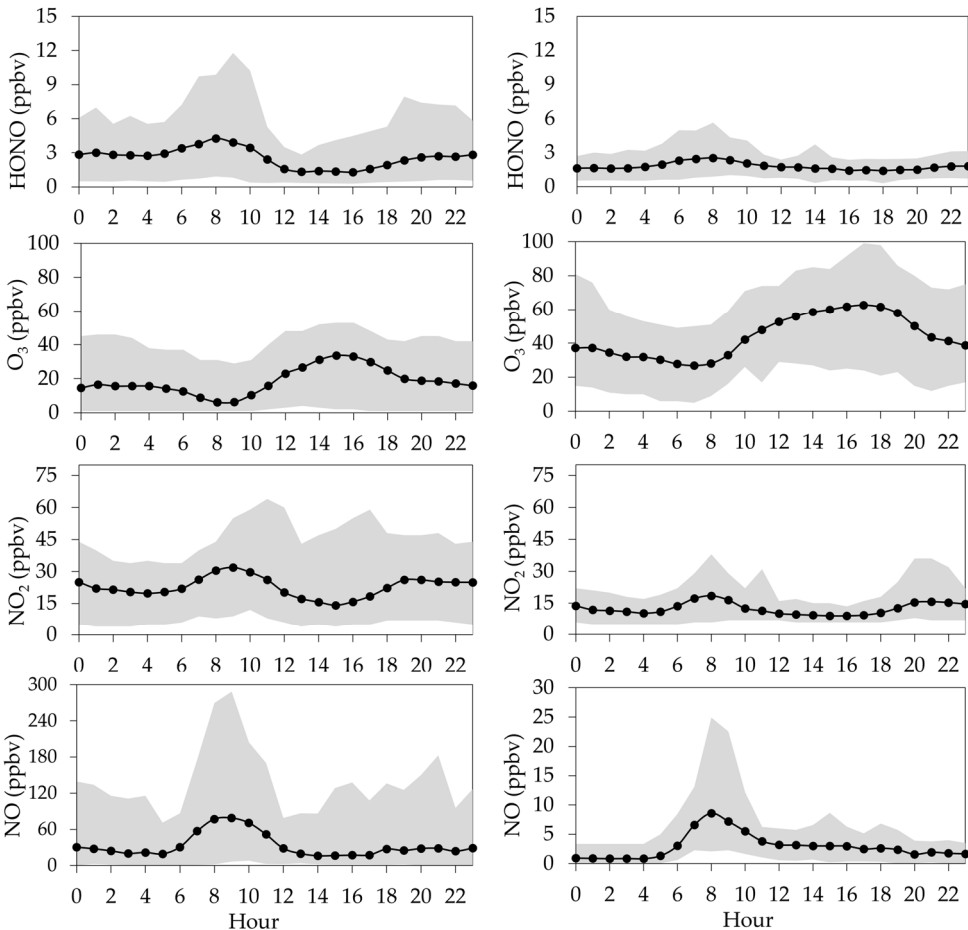

**Figure 5.** Diurnal patterns of precursor gases representing winter (**left**) and summer (**right**). The gray area represents the range (standard deviation) of hourly averaged data.

### 3.2. Zero-Dimensional Box Model

In this study, the F0AM model was utilized to analyze the specific chemical reactions of HONO and HNO$_3$, focusing on the summer season when photochemical reactions particularly active. As substances such as OH radicals, HO$_2$, NO$_3$, N$_2$O$_5$, and PAN were not the targeted measurements, a 0-D box model, namely the F0AM model, was employed to quantitatively identify the concentrations of these substances present in the atmosphere during the measurement period [25,28]. Incorporating the chemical reaction mechanism of MCM v3.1.1, the model accounts for a comprehensive set of reactions governing the behavior of HONO and HNO$_3$. MCM v3.1.1 provides a detailed framework for understanding the intricacies of atmospheric chemistry, assisting in the quantitative analysis of these compounds. The applied data, encompassing hourly averaged datasets (including meteorological data), further enhances the model's ability to simulate and interpret the atmospheric processes during the study period.

To acquire VOC data for simulating the F0AM model in this study, VOC measurements were conducted using a PTR-ToF-MS (IONICON) at the same measurement site. The collected data were then applied to the F0AM model. The selection of VOC substances for model calculations were based on their measured concentrations and maximum incremental reactivity (MIR) values. MIR is a metric developed by the California Air Resource Board that quantitatively assesses the impact of VOCs on ground-level O$_3$ [34–36]. Substances with a product of the overall average concentration and the MIR value of the measured VOC substances ≥ 1 were chosen and incorporated into the model. The selected VOC species are presented in Table 3.

**Table 3.** VOCs used for the F0AM model measured using a PTR-ToF-MS.

|  | VOC Species | Conc. (ppbv) | MIR | Conc. $\times$ MIR |
|---|---|---|---|---|
| 1 | Propene | 3.7 | 11.7 | 42.6 |
| 2 | Butene | 2.8 | 9.7 | 27.4 |
| 3 | Butanol | 2.1 | 6.0 | 12.5 |
| 4 | m-Xylene | 1.3 | 9.8 | 12.3 |
| 5 | Acetaldehyde | 1.6 | 6.5 | 10.2 |
| 6 | Formaldehyde | 0.6 | 9.5 | 5.3 |
| 7 | Methanol | 7.3 | 0.7 | 4.9 |
| 8 | Toluene | 1.2 | 4.0 | 4.8 |
| 9 | Ethanol | 2.9 | 1.5 | 4.4 |
| 10 | 1,2,4-Trimethylbenzene | 0.5 | 8.9 | 4.2 |
| 11 | 1,3-Butadiene | 0.3 | 12.6 | 4.0 |
| 12 | Ethene | 0.4 | 9.0 | 4.0 |
| 13 | Isoprene | 0.4 | 10.6 | 3.9 |
| 14 | Crotonaldehyde | 0.4 | 9.4 | 3.8 |
| 15 | Acrolein | 0.5 | 7.5 | 3.4 |
| 16 | n-Hexane | 1.8 | 1.2 | 2.2 |
| 17 | Ethylbenzene | 0.6 | 3.0 | 1.9 |
| 18 | iso-Butyl alcohol | 0.8 | 2.5 | 1.9 |
| 19 | a-Pinene | 0.4 | 4.5 | 1.7 |
| 20 | n-Valeraldehyde | 0.3 | 5.1 | 1.7 |
| 21 | Acetic acid | 2.2 | 0.7 | 1.5 |
| 22 | 2-Ethoxyethanol | 0.4 | 3.7 | 1.4 |
| 23 | Methyl iso-butyl ketone | 0.3 | 3.9 | 1.3 |
| 24 | 2-Methoxyethanol | 0.4 | 2.9 | 1.2 |
| 25 | Acetone | 3.1 | 0.4 | 1.1 |

The model employed in this study simulated the concentrations of various precursors in an actual atmospheric environment. Table 4 represents a comparison between the observed concentrations of the precursors and air pollutants and the concentrations simulated using the model. Root mean square deviation (RMSD) served as an index to assess the accuracy of the model estimates. The smaller RMSD value indicates a better agreement between the model calculations and the measured value. The RMSD value was generally 1 or less for most VOCs, except formaldehyde. Inorganic species like HONO, $O_3$, and $NO_X$, exhibited relatively high RMSD values, highlighting a limitation of the F0AM model designed for the photochemical simulation of VOC species.

**Table 4.** Comparison between measured and modeled values of observed gases.

| Species | Measured Conc. | | Modeled Conc. | | RMSD (ppbv) |
|---|---|---|---|---|---|
|  | AVG | STD | AVG | STD | |
| HONO | 1.79 | 0.76 | 0.85 | 0.36 | 1.02 |
| $HNO_3$ | 0.10 | 0.03 | 0.76 | 0.28 | 0.71 |
| $O_3$ | 43.8 | 18.3 | 45.0 | 18.1 | 2.91 |
| $NO_2$ | 12.5 | 5.51 | 13.7 | 5.49 | 2.45 |
| NO | 2.33 | 2.58 | 0.83 | 1.08 | 2.57 |
| CO | 523 | 94.2 | 523 | 94.3 | 0.30 |
| $SO_2$ | 2.80 | 1.07 | 2.78 | 1.06 | 0.01 |
| Propene | 3.64 | 1.74 | 3.04 | 1.43 | 0.69 |
| Butene | 2.82 | 1.13 | 2.33 | 0.94 | 0.53 |
| Butanol | 2.10 | 1.17 | 2.02 | 1.13 | 0.10 |
| m-Xylene | 1.26 | 0.74 | 1.13 | 0.66 | 0.16 |
| Acetaldehyde | 1.56 | 0.93 | 2.03 | 1.05 | 0.54 |

**Table 4.** *Cont.*

| Species | Measured Conc. | | Modeled Conc. | | RMSD (ppbv) |
|---|---|---|---|---|---|
| | AVG | STD | AVG | STD | |
| Formaldehyde | 0.56 | 0.25 | 1.85 | 0.63 | 1.38 |
| Methanol | 7.31 | 2.40 | 7.28 | 2.40 | 0.03 |
| Toluene | 1.19 | 0.83 | 1.15 | 0.81 | 0.04 |
| Ethanol | 2.89 | 2.22 | 2.85 | 2.19 | 0.06 |
| 1,2,4-Trimethylbenzene | 0.47 | 0.24 | 0.41 | 0.20 | 0.08 |
| 1,3-Butadiene | 0.32 | 0.13 | 0.22 | 0.09 | 0.11 |
| Ethene | 0.42 | 0.21 | 0.40 | 0.20 | 0.02 |
| Isoprene | 0.37 | 0.10 | 0.20 | 0.06 | 0.18 |
| Crotonaldehyde | 0.40 | 0.13 | 0.34 | 0.11 | 0.07 |
| Acrolein | 0.46 | 0.21 | 0.48 | 0.19 | 0.04 |
| n-Hexane | 1.78 | 0.98 | 1.73 | 0.95 | 0.06 |
| Ethylbenzene | 0.64 | 0.34 | 0.62 | 0.32 | 0.02 |
| iso-butyl alcohol | 0.75 | 0.33 | 0.72 | 0.31 | 0.04 |
| a-Pinene | 0.38 | 0.19 | 0.13 | 0.15 | 0.26 |
| n-Valeraldehyde | 0.33 | 0.09 | 0.28 | 0.08 | 0.04 |
| Acetic acid | 2.21 | 1.09 | 2.21 | 1.09 | 0.01 |
| 2-Ethoxyethanol | 0.36 | 0.14 | 0.33 | 0.13 | 0.03 |
| Methyl iso-butyl ketone | 0.33 | 0.13 | 0.31 | 0.13 | 0.02 |
| 2-Methoxyethanol | 0.40 | 0.15 | 0.38 | 0.14 | 0.03 |
| Acetone | 3.08 | 1.09 | 3.14 | 1.09 | 0.06 |

*3.3. Simulated $HO_X$ and $NO_Z$ Species*

$HO_X$ is a term collectively encompassing the OH and $HO_2$ radicals, which play a significant oxidizing role against VOCs, $NO_X$, $O_3$, and other atmospheric substances [35,36]. The quantities of $NO_y$ substances, including $N_2O_5$, $NO_3$, and PAN, were calculated concurrently with those of the $HO_X$ radicals. $N_2O_5$ and $NO_3$ are nitrogen oxides intricately involved in the formation of $HNO_3$, while PAN is a nitrogen oxides component contributing to photochemical smog [3,4,15].

The average OH and $HO_2$ concentrations in the summer atmosphere, calculated using the F0AM model, were $1.1 \pm 0.25 \times 10^6$ molecules cm$^{-3}$ (n = 687) and $1.4 \pm 0.95 \times 10^8$ molecules cm$^{-3}$ (n = 687), respectively. The OH radical concentrations are simulated at a lower level than the actual measured values in the other research [37–39]. However, according to previous research that used modeling techniques to calculate global OH radical concentrations, the reported average range of OH concentrations is $5.6$–$14.6 \times 10^5$ molecules cm$^{-3}$ [40]. Therefore, the simulated OH concentrations in this study are reasonably calculated in comparison. The $N_2O_5$, $NO_3$, and PAN concentrations in the atmosphere, calculated using the model, were $17 \pm 9$ pptv (n = 687), $1.6 \pm 1.5$ pptv (n = 687), and $0.13 \pm 0.07$ ppbv (n = 687), respectively. Figure 6 shows the composition ratio of $NO_X$ to $NO_Z$ and the $NO_Z$ species (HONO, $HNO_3$, $N_2O_5$, $NO_3$, and PAN). Among $NO_y$, $NO_2$ accounted for the largest proportion at 71.7%, followed by NO and $NO_Z$ at 16.8% and 11.7%, respectively. For $NO_Z$, HONO accounted for 87.6% of the total. $N_2O_5$, $NO_3$, and PAN accounted for only 7.3% of $NO_Z$. The $HNO_3$ concentration in the atmosphere is significantly lower than that of HONO.

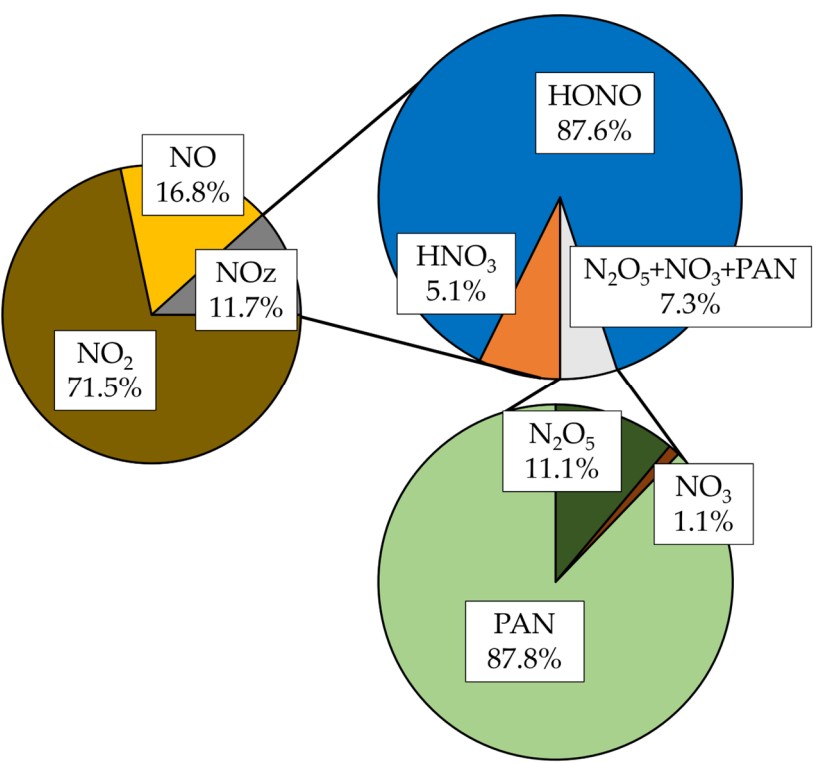

**Figure 6.** The distribution ratio of $NO_y$. NO, $NO_2$, HONO, and $HNO_3$ concentrations were measured and $NO_3$, $N_2O_5$, PAN concentrations were calculated using the F0AM model.

*3.4. Mechanisms of HONO Production and Conversion*

Despite recent studies on the production and loss mechanisms of *HONO*, there are still unidentified sources of *HONO* emissions [8,11,12]. Therefore, it was necessary to incorporate an additional unknown source of *HONO* emissions ($P_{unknown}$) in this study. This source was estimated based on the variations in the measured *HONO* concentration over time, attributed to several reactions, including the reaction between *OH* radicals and *NO* and the photolysis of *HONO*.

$$\frac{\delta[HONO]}{\delta t} = \left(P_{NO+OH} + P_{2NO_2+H_2O} + P_{emis} + P_{het} + P_{unknown}\right) - \left(L_{2HONO} + L_{HONO+OH} + L_{photo}\right)$$

Then,

$$P_{unknown} = \frac{\delta[HONO]}{\delta t} - \left(P_{NO+OH} + P_{2NO_2+H_2O} + P_{emis} + P_{het}\right) + \left(L_{2HONO} + L_{HONO+OH} + L_{photo}\right)$$

$$P_{NO+OH} = k_{NO+OH}[NO][OH]$$

$$P_{2NO_2+H_2O} = k_{2NO_2+H_2O}[NO_2]^2[H_2O]$$

$$P_{direct} = [NO_X] \times 0.0065$$

$$P_{het} = C_{HONO}[NO_2]$$

$$L_{2HONO} = k_{2HONO}[HONO]^2$$

$$L_{HONO+OH} = k_{HONO+OH}[HONO]$$

$$L_{photo} = J_{HONO}[HONO]$$

$\delta[HONO]/\delta t$ represents the *HONO* concentration of change over time (ppbv hr$^{-1}$). *P* and *L* with the reactants written in subscripts represent the budget (in ppbv hr$^{-1}$) of *HONO* produced or lost (i.e., sinks), respectively. $P_{direct}$ corresponds to the direct emission rate of *HONO* from automobile engine combustion [41].

$P_{het}$ refers to the *HONO* concentration converted from $NO_2$ through heterogeneous reactions and can be estimated by multiplying the $C_{HONO}$ and $NO_2$ concentrations [42]. The $C_{HONO}$ for calculating the budget of *HONO* converted from $NO_2$ can be calculated as in the following equation [43,44]:

$$C_{HONO} = \frac{[HONO]_{t_2} - [HONO]_{t_1}}{[NO_2] \times (t_2 - t_1)}$$

Research on $C_{HONO}$, a coefficient for calculating the budget of *HONO* converted from $NO_2$, has been continuously conducted, and the average value of $C_{HONO}$ calculated in this study was $0.011 \pm 0.021$ hr$^{-1}$, which did not differ substantially from that in previous studies [12,41,42]. $J_{HONO}$ represents the numerical coefficient of the fractional photolysis of *HONO* and is expressed in reciprocal seconds (s$^{-1}$). As *J* is dependent on the solar zenith angle (SZA) at the time of measurement, its application is limited to daylight hours. In this study, the average $J_{HONO}$ value was 0.0019 s$^{-1}$, with the maximum recorded value during the measurement period reaching 0.0024 s$^{-1}$.

In the case of the mechanisms leading to the decrease in *HONO*, $L_{photo}$ corresponded to the decrease through photolysis, whereas diffusion or vertical/horizontal physical transport and deposition were not considered due to their minimal contributions.

Changes in the HONO concentration over time were quantified by multiplying the reaction rate coefficients corresponding to each reaction by the concentrations of the reactants. Table 5 summarizes the reactions and reaction rate coefficients of HONO used in this study.

**Table 5.** Summary of the reaction rate constants of HONO.

| | Reaction | Constant | Unit | Reference |
|---|---|---|---|---|
| $P_{NO+OH}$ | $NO + OH \rightarrow HONO$ | $7.31 \times 10^{-12}$ | cm$^3$ mole$^{-1}$ s$^{-1}$ | [45] |
| $P_{2NO_2+H_2O}$ | $2NO_2 + H_2O \rightarrow HONO + HNO_3$ | $1.47 \times 10^{-23}$ | L$^2$ mol$^{-2}$ s$^{-1}$ | [46] |
| $P_{direct}$ | Depends on concentration of $NO_X$ | $[NO_X] \times 0.0065$ | - (a) | [47] |
| $P_{het}$ | Depends on conversion rate from $NO_2$ | $C_{HONO} \times [NO_2]$ | - | [41,42] |
| $L_{photo}$ | $HONO + hv \rightarrow NO + OH$ | $J_{HONO} \times [HONO]$ | - (a) | [47] |
| $L_{2HONO}$ | $2HONO \rightarrow NO + NO_2 + H_2O$ | $1.4 \times 10^{-3}$ | ppm$^{-1}$ min$^{-1}$ | [48] |
| $L_{HONO+OH}$ | $HONO + OH \rightarrow NO_2 + H_2O$ | $5.59 \times 10^{-11}$ | cm$^3$ mole$^{-1}$ s$^{-1}$ | [48] |

(a) The unit must be converted to ppb/h.

Figure 7 illustrates the calculated time-dependent changes in HONO concentration and the processes of production, conversion, and loss of HONO using reaction coefficients. The results revealed that the unknown source of HONO emissions ($P_{unknown}$) constituted 55.7% of the total HONO production. Among the various HONO production processes, the heterogeneous reaction pathway ($P_{het}$, 21.3%) was the most dominant, excluding the unknown source. The direct emissions (vehicle exhaust from engine combustion) accounted for 7.9%. The combined production from the two considered reaction equations was approximately 15%.

Regarding the loss of HONO, photolysis of HONO ($L_{photo}$) was the dominant mechanism, constituting 77.7%, followed by $L_{HONO+OH}$ at 22.3%. The reduction in the HONO budget due to $L_{2HONO}$ was negligible.

Figure 8 illustrates the diurnal variations in HONO production and loss. HONO production exhibited a significant increase in the morning, with the primary source identified as an unknown emission source. During the morning rush hour, the contribution of direct emissions increased due to elevated $NO_X$ emissions. However, after sunset, the contribution of the heterogeneous reaction pathway ($P_{het}$) significantly increased.

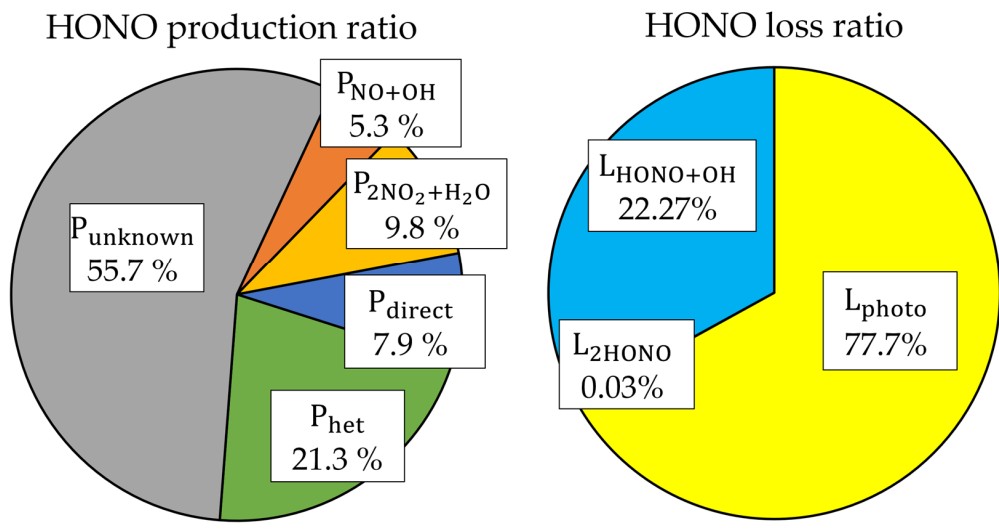

**Figure 7.** Composition ratio of productions and losses of HONO.

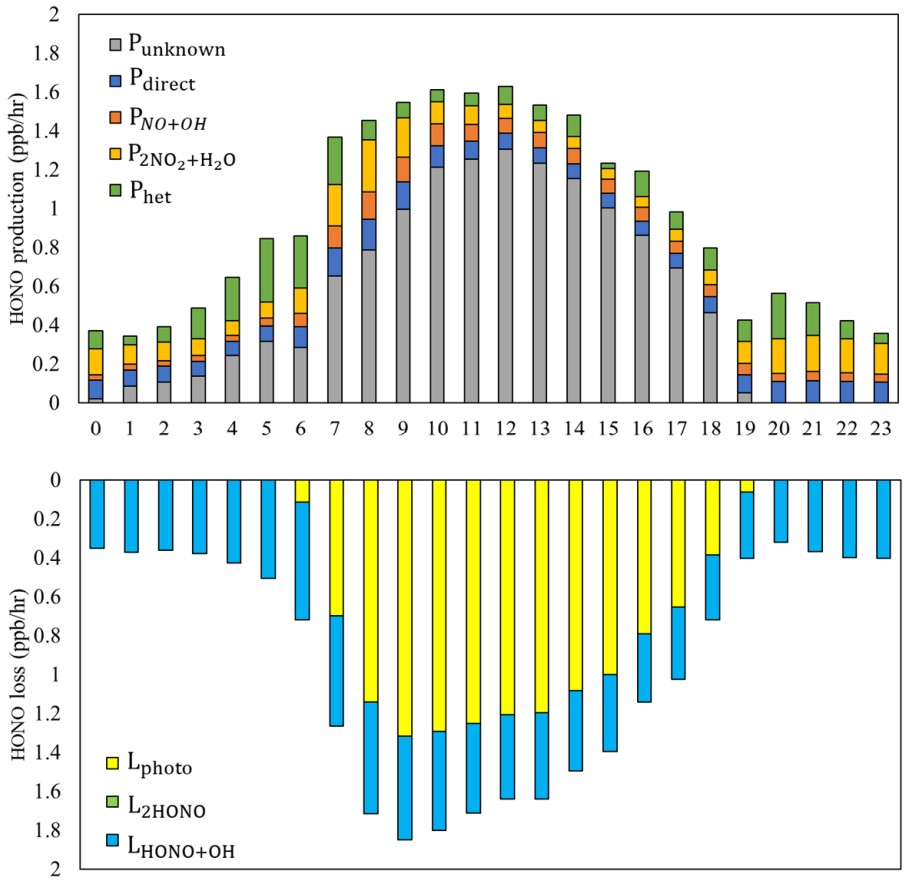

**Figure 8.** Diurnal variation in the production and loss rate of HONO.

HONO loss exhibited an increasing trend in photolysis-induced HONO loss after sunrise; however, the reduction rate decreased over time as the SZA decreased. The second-largest loss was attributed to the gas-phase reaction between HONO and OH radicals ($L_{HONO+OH}$), where the OH radicals produced from the photolysis of HONO reacted with another HONO molecule, leading to the reduction in HONO. This reaction is presumed to be dominant at night, contributing to the decrease in atmospheric HONO concentration.

### 3.5. Mechanisms of HNO₃ Production and Conversion

$HNO_3$ is produced through various pathways, some of which are more important than others. However, the decrease in $HNO_3$ is primarily due to heterogeneous reactions (gas-to-liquid) rather than homogeneous reactions between the gases [18,49]. The most important mechanism for this decrease is the aerosolization reaction, in which $NH_4NO_3$ is produced from the reaction between ammonia in the atmosphere, and $HNO_3$ is converted into aerosols [49].

In this study, the change in $HNO_3$ gas concentration over time was estimated using various mechanisms. The remaining change in concentration was attributed to the amount of $HNO_3$ that decreased due to aerosolization, as well as through dry and wet deposition.

$$\frac{\delta[HNO_3]}{\delta t} = \left( P_{NO_2+OH} + P_{N_2O_5+H_2O} + P_{NO_3+HO_2} + P_{2NO_2+H_2O}^{HNO_3} \right) \\ - \left( L_{photo}^{HNO_3} + L_{HNO_3+OH} + L_{aero+dep} \right)$$

Then,

$$L_{aero+dep} = \left( P_{NO_2+OH} + P_{N_2O_5+H_2O} + P_{NO_3+HO_2} + P_{2NO_2+H_2O}^{HNO_3} \right) \\ - \left( \frac{\delta[HNO_3]}{\delta t} + L_{photo}^{HNO_3} + L_{HNO_3+OH} \right)$$

$$P_{NO_2+OH} = k_{NO_2+OH}[NO_2][OH]$$

$$P_{N_2O_5+H_2O} = k_{N_2O_5+H_2O}[N_2O_5][H_2O]$$

$$P_{NO_3+HO_2} = k_{NO_3+HO_2}[NO_3][HO_2]$$

$$P_{2NO_2+H_2O}^{HNO_3} = k_{2NO_2+H_2O}[NO_2]^2[H_2O]$$

$$L_{HNO_3+OH} = k_{HNO_3+OH}[HNO_3][OH]$$

$\delta[HNO_3]/\delta t$ represents the quantity of change in $HNO_3$ concentration per unit time (ppb hr$^{-1}$), and $P$ and $L$ represent the budget (in ppb hr$^{-1}$) of $HNO_3$ that is produced or lost through reactions as several equations summarized in Table 6. Similarly to the calculation of $HONO$, these equations were used to estimate the differences in $HNO_3$ concentration over time.

**Table 6.** Summary of constants for reaction rate of HNO₃.

| | Reaction | Constant | Unit | Reference |
|---|---|---|---|---|
| $P_{NO_2+OH}$ | $NO_2 + OH \rightarrow HNO_3$ | $1.051 \times 10^{-11}$ | cm³ molecules$^{-1}$ s$^{-1}$ | [45] |
| $P_{N_2O_5+H_2O}$ | $N_2O_5 + H_2O \rightarrow 2HNO_3$ | $2.5 \times 10^{-22}$ | cm³ molecules$^{-1}$ s$^{-1}$ | [45] |
| $P_{NO_3+HO_2}$ | $NO_3 + HO_2 \rightarrow HNO_3 + O_2$ | $1.9 \times 10^{-12}$ | cm³ molecules$^{-1}$ s$^{-1}$ | [16] |
| $P_{2NO_2+H_2O}^{HNO_3}$ | $2NO_2 + H_2O \rightarrow HONO + HNO_3$ | $5.5 \times 10^4$ | L² mol$^{-2}$ s$^{-1}$ | [46] |
| $L_{photo}^{HNO_3}$ | $HNO_3 + hv \rightarrow NO_2 + OH$ | $J_{HNO_3} \times [HNO_3]$ | - [a] | [47] |
| $L_{HNO_3+OH}$ | $HNO_3 + OH \rightarrow H_2O + NO_3$ | $1.51 \times 10^{-13}$ | cm³ molecules$^{-1}$ s$^{-1}$ | [45] |

[a] The unit must be converted to ppb hr$^{-1}$.

Figure 9 depicts the ratios of HNO₃ production, conversion, and loss rates based on the reaction coefficients. Concerning production mechanisms, HNO₃ production via the reaction of $NO_2$ and $OH$ ($P_{NO_2+OH}$) accounted for approximately 41.0%. The reaction between $N_2O_5$ and $H_2O$, occurring during the nighttime denoted as $P_{N_2O_5+H_2O}$, contributed significantly with 49.4%, indicating its dominance in total HNO₃ production. $P_{2NO_2+H_2O}^{HNO_3}$, representing the hydrolysis reaction with $NO_2$, accounted for 9.4%. Meanwhile, the production from the reaction of $NO_3$ and $HO_2$ ($P_{NO_3+HO_2}$) had a minor composition ratio.

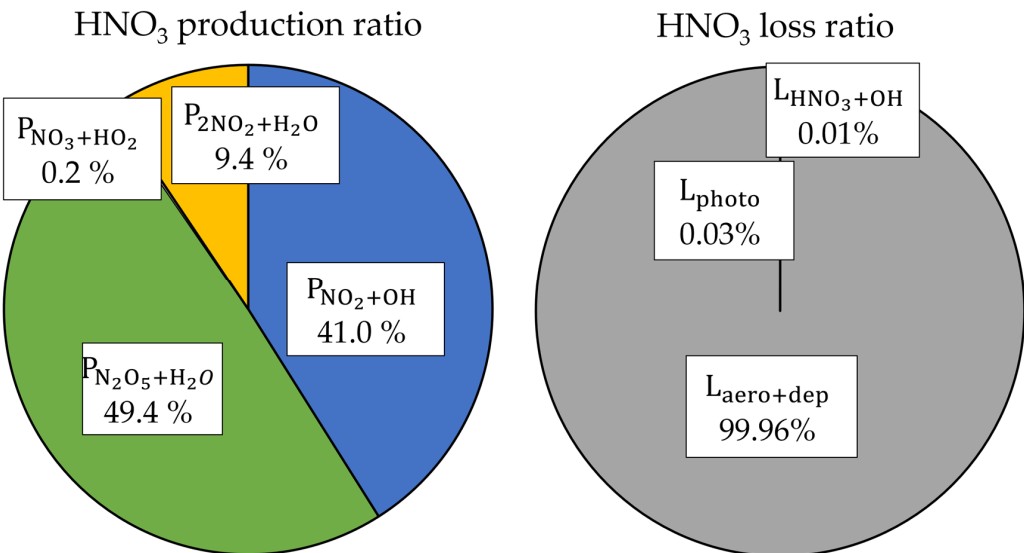

**Figure 9.** Composition ratio of productions and losses of $HNO_3$.

In the case of loss rates, the majority $HNO_3$ underwent reduction through aerosolization or deposition, serving as the primary causes of reduction. Further research is required to elucidate the relationship between aerosolization pathways and deposition.

Figure 10 illustrates the diurnal variations in $HNO_3$ production over time. From 7 to 8 a.m., the production rate of $HNO_3$ by $P_{NO_2+OH}$ was high. The quantity of $HNO_3$ produced by gas-phase reactions between $NO_2$ and $OH$ ($P_{NO_2+OH}$) decreased during the day, while the production rate of $HNO_3$ formed by the reaction between $N_2O_5$ and $H_2O$ ($P_{N_2O_5+H_2O}$) increased. Because of the reaction of $P_{2NO_2+H_2O}^{HNO_3}$, the production rate of $HNO_3$ was determined by the same equivalent ratio as that of HONO, and it was relatively small compared to the production rates by other reactions. The production rate of $P_{NO_3+HO_2}$ was low.

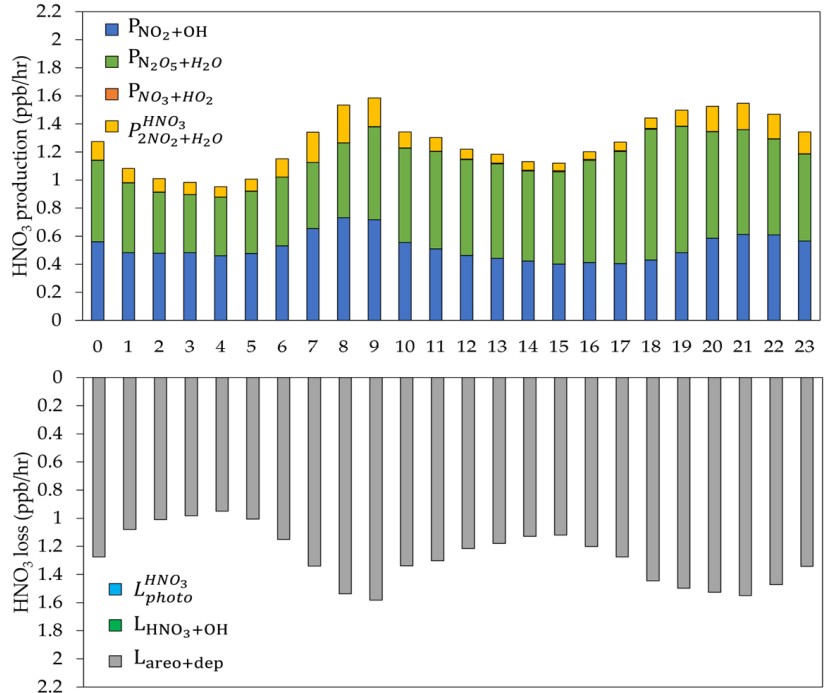

**Figure 10.** Diurnal variation in production and loss rates of $HNO_3$.

Regarding the reduction in $HNO_3$, the conversion rate to aerosols was proportional to the $HNO_3$ concentration. The reduction by aerosol conversion and deposition surpassed that by photolysis or reaction with OH radicals, making it challenging to analyze the one-to-one reduction. Conducting further research using the thermodynamic equilibrium model from gas to particles could provide more detailed estimates of the reduction rate in $HNO_3$ [50].

To evaluate the reliability of $L_{aero+dep}$, quantified from various reaction equations and the concentration budget over time, a quantitative comparison was conducted between the model-simulated and observed values. Since the F0AM model does not account for particle transformation mechanisms or dry and wet deposition processes, the $HNO_3$ concentration estimated by the F0AM model was higher than the actual measured value. Hence, the difference between the observed $HNO_3$ concentration values and those simulated by the model was assumed to be aerosolized or deposited. A comparative analysis of these two indicators was then performed.

As depicted in Figure 11, the estimated concentration of $HNO_3$ transformed into particles and deposited exhibited a relatively similar trend to the $PM_{2.5}$ concentration, except for the peak case during the measurement. However, the trend of loss through the pathway of aerosolization and deposition did not align with that of $GAP_{HNO_3}$. Figure 12 illustrates the diurnal patterns of the $HNO_3$ budget for $L_{aero+dep}$ and $GAP_{HNO_3}$, showing a difference in the values of the two indicators; nevertheless, the pattern itself was similar.

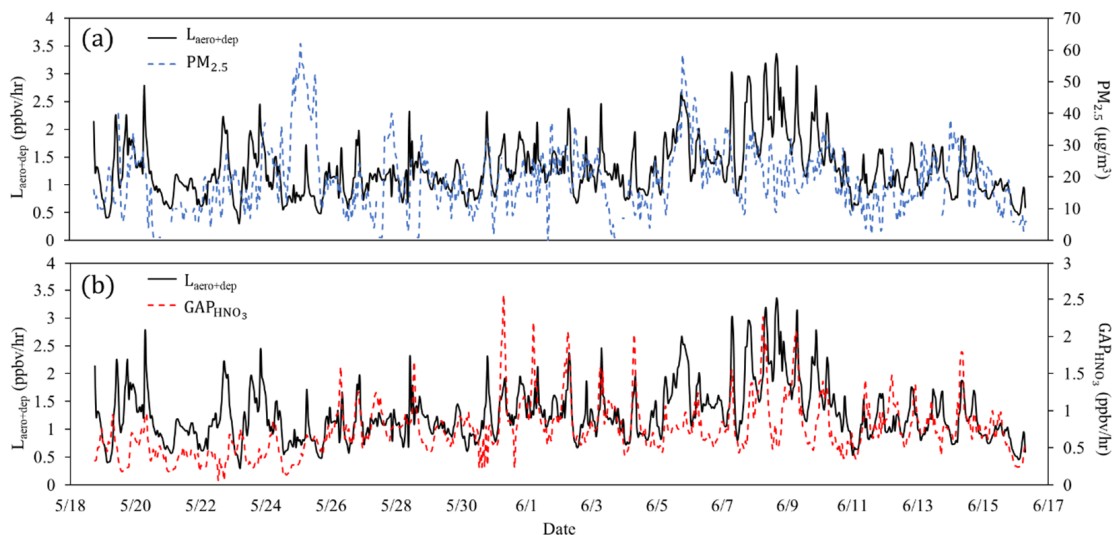

**Figure 11.** Time-series distribution comparing $L_{aero+dep}$ with two indicators: (**a**) with $PM_{2.5}$ (the blue dotted line); (**b**) with $GAP_{HNO_3}$ (the red dotted line).

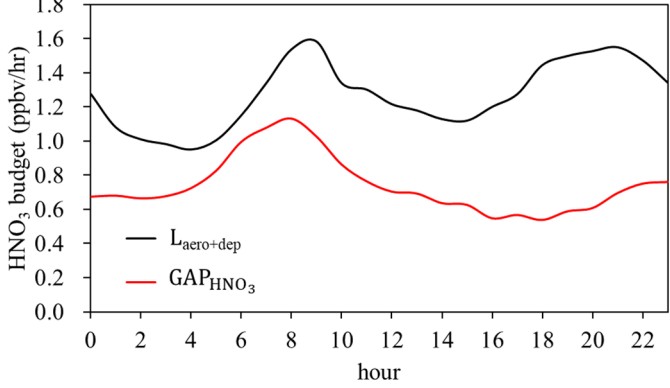

**Figure 12.** Diurnal pattens of $L_{aero+dep}$ and $GAP_{HNO_3}$.

The $L_{aero+dep}$ value calculated in this study is an estimated parameter; this does not imply that the absolute budget of $HNO_3$ is necessarily transformed into an aerosol. To determine the conversion to aerosols in detail, it was necessary to employ a gas–particle equilibrium model to quantify the values that vary depending on changes in conditions such as temperature and humidity [50]. It was suggested that a comparative study on the concentration of $NO_3^-$ ions ($\mu g\ m^{-3}$) obtained by analyzing the components of $PM_{2.5}$ in the atmosphere and the budget converted from $HNO_3$ is required.

## 4. Conclusions

This study conducted measurements of HONO and $HNO_3$ concentrations, along with their precursor gases, in the atmosphere during both winter and summer in Daejeon, Korea. The average HONO concentrations measured were found to be $2.59 \pm 1.91$ ppbv in winter and $1.79 \pm 0.76$ ppbv in summer, while the $HNO_3$ concentrations were $0.72 \pm 0.61$ ppbv in winter and $0.1 \pm 0.03$ ppbv in summer. In particular, the HONO concentration exhibited a decrease at sunrise through photolysis, contributing to the production of OH radicals and influencing the accumulation of $O_3$ in the atmosphere. Analyses were conducted on changes in the $HONO/NO_Z$ and $HNO_3/NO_Z$ ratios based on the composition of $NO_y$ substances and the measurement period. Utilizing the 0-D box model, concentrations of $N_2O_5$, $NO_3$, and PAN were calculated.

Quantitative evaluations of production and conversion mechanisms were conducted using observed and simulated F0AM concentration data, employing several reaction equations and constants for HONO and $HNO_3$. The HONO production through the heterogeneous reaction of $NO_2$ ($P_{het}$) accounted for 21.3%, with an estimated proportion of 55.7% for unknown source ($P_{unknown}$). Further detailed research is required to determine the budget of HONO with an unknown source. Regarding HONO reduction, the largest loss occurred through photolysis during the day (77.7%), with remaining reduction due to the reaction of high OH radicals with HONO.

For $HNO_3$ production, the majority involved the reaction of $N_2O_5$ and $H_2O$ (49.4%), with 41.0% produced by the homogeneous reaction of $NO_2$ and OH radicals. In terms of $HNO_3$ budget loss, 99.9% of the reduction was converted to aerosols or deposited. A comparison between the F0AM model result values and measured $HNO_3$ concentrations ($GAP_{HNO_3}$) and $L_{aero+dep}$ revealed similar diurnal patterns over time.

**Author Contributions:** Conceptualization, J.H.; methodology, J.E.; validation, J.H. and J.E.; formal analysis, K.K., C.L., D.C. and S.H.; investigation, K.K. and D.C.; data curation, J.H. and J.E.; writing—original draft preparation, K.K.; writing—review and editing, K.K., C.L. and S.H.; supervision, J.H.; funding acquisition, J.H. All authors have read and agreed to the published version of the manuscript.

**Funding:** This research was supported by the FRIEND (Fine Particle Research Initiative in East Asia Considering National Differences) Project through the National Research Foundation of Korea (NRF) funded by the Ministry of Science and ICT (2020M3G1A1114999) and Experts Training Graduate Program for Particulate Matter Management from the Ministry of Environment, Korea.

**Institutional Review Board Statement:** Not applicable.

**Informed Consent Statement:** Not applicable.

**Data Availability Statement:** The data presented in this study are available on request from the corresponding author.

**Conflicts of Interest:** Sangwoo Han is employee of E2M3 Inc., Anyang 14059, Republic of Korea. The paper reflects the views of the scientists and not the company.

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
