# Peer review of "A Study on the Formation Reactions and Conversion Mechanisms of HONO and HNO3 in the Atmosphere of Daejeon, Korea"

_atmosphere, doi:10.3390/atmos15030267_

Round 1

Reviewer 1 Report

Comments and Suggestions for Authors

In this study, the authors analyzed the formation and conversion processes of HONO and HNO3 and their implication in atmospheric chemistry and gas-to-particle conversion. Their work clarifies some missing sources for these species and the importance of their inclusion in chemical models. Despite some language challenges, the manuscript is well written and results discussion is sound. However, the following points have to be addressed prior to publication.

Abstract:

a) Nitrogen oxides (NOx) stands for NO+NO2 and therefore should generally be in the plural form.

b) Improve the sentence that presents the method used.

Introduction:

a) The first paragraph is somewhat elementary and the verbs tenses are used randomly. The authors are encouraged to make some improvements.

b) Line 63-65. Can the authors elaborate more on what they mean by case day and non-case days?

c) Line 67-69. Check that the sentence is correctly written and that nothing is missing.

d) Line 76. Change “detailed materials” to “individual species”

e) Line 85. Change “measurements” to “measurement results”

Materials and Methods:

a) Line 115. It would be good if the authors could define PPDS-IC at the first use and also check other similar usages throughout the manuscript

Results:

a) Table 2. The caption should come before the table and not the other way around.

b) Line 196-196. Should be “…measured mixing ratios of …were relatively low.”

Fig.6. NO, NO2, HONO, and HNO3 concentrations were measured and NO3, N2O5, PAN concentrations were calculated…

Make similar corrections at line 428

c) Lines 181 and 182: Use “higher than” instead of “greater than”. Check similar usages elsewhere in the manuscript

d) Can the authors elaborate more on the aerosolization process of HNO3? Is HNO3 in gaseous or liquid state?

Other general comments

a) The authors have repeatedly used the term “aerosolization” as a major mechanism for the reduction of HNO3 concentration. Knowing that aerosolization is the act of forming a spray from a liquid or a particulate substance, can they elaborate more on it?

b) Since the derivation in equations and elsewhere in the text is taken solely relative to time, can the authors use the total derivative instead of the partial derivative?

b) The language has to be edited.

Comments on the Quality of English Language

The manuscript needs English language editing. General grammar and readability should be improved

Author Response

Dear Reviewers,

We sincerely appreciate the time and effort you dedicated to reviewing our manuscript. Your insightful comments and constructive feedback have been invaluable in enhancing the quality of our work. We want to express our gratitude for your thorough examination of our paper and for your thoughtful suggestions.

We want to assure you that we have carefully considered each of your comments and strived to incorporate them effectively into the revised manuscript. Our detailed notes can be found in the attached Word document. We believe that these clarifications and adjustments have significantly improved the overall clarity and quality of the paper.

Your expertise and thoughtful consideration have contributed immensely to the refinement of our research. We are sincerely thankful for your dedication to maintaining the scholarly standards of our work. We hope that the modifications made in response to your feedback meet your expectations.

Thank you once again for your time, commitment, and valuable contributions to our manuscript.

Best regards,
Kim.

Reviewer 2 Report

Comments and Suggestions for Authors

This study observed gaseous HONO and HNO3 using IC. And a box model is utilized to analyze formation and loss pathways of HONO and HNO3. There are some issues about this article and significant improvements should be made.

1.       The abstract should mainly describe the research results, not the importance of the study.

2.       Are particulate nitrates and nitrites absorbed in the solution? And do they interfere with detection? Is this detection method compared with other instruments, such as optical spectroscopic methods and long-path absorption photometric technique?

3.       In lines168-170, how far is the National Institute of Environment and Research from the observation site?

4.       In Lines 193-197, the description of pollutants are not exact. And what conclusion is this section trying to illustrate? In other words, what is behind the phenomenon?

5.       In Lines 233-235, how does the model work? It needs to be explained in detail.

6.       In Lines 250-251, the simulated OH concentration is 1.1 ×106 molecule/cm3, did the author compared it with previous studies? As far as I know, the concentration of OH radicals should be higher in the summer, which would cause inaccuracies in the HONO simulations below.

7.       The model simulation didn’t consider dry deposition which is a significant sink in the nighttime.

8.       The calculation of CHONO needs to be explained in detail.

Comments on the Quality of English Language

Language needs further improvement and embellishment.

Author Response

Dear Reviewer,

We sincerely appreciate the time and effort you dedicated to reviewing our manuscript. Your insightful comments and constructive feedback have been invaluable in enhancing the quality of our work. We want to express our gratitude for your thorough examination of our paper and for your thoughtful suggestions.

We want to assure you that we have carefully considered each of your comments and strived to incorporate them effectively into the revised manuscript. Our detailed notes can be found in the attached Word document. We believe that these clarifications and adjustments have significantly improved the overall clarity and quality of the paper.

Your expertise and thoughtful consideration have contributed immensely to the refinement of our research. We are sincerely thankful for your dedication to maintaining the scholarly standards of our work. We hope that the modifications made in response to your feedback meet your expectations.

Thank you once again for your time, commitment, and valuable contributions to our manuscript.

Best regards,
Kim.

Reviewer 3 Report

Comments and Suggestions for Authors

This paper employ field measurement technique to study formation reaction and mechanism of HONO and HNO3 in the atmosphere of Daejeon in Korea. 

while the paper did not give appropriate details of the conversion mechanism, the title and body not in agreement. The introduction section need more recent papers to substantiate results. The data and method section is not well presented. There is no detail information of measurement and the discussion of why seasons is missing. what data were actually been measured during field measurement?

more details is required in this section. There are several subsections with limited information and should be improved. The figures need to be discussed in more scientific way and the conclusion well presented. The abstract need total rewrite to present clearly the objectives of study. In this present form, I do not recommend the paper for publication.   

Comments on the Quality of English Language

Need to be improved. 

Author Response

(The authors gave the same response as above.)

Reviewer 4 Report

Comments and Suggestions for Authors

This study shows the results of measurements of the concentration of HONO and HNO3 and precursor gases in the atmosphere in winter and summer in Daejeon, Korea. The average values of the measured concentrations of HONO and HNO3 are shown and the changes in the ratios of HONO/NOZ and HNO3/NOZ are analyzed. The paper presents quantitative estimates of the mechanisms of production and conversion of HONO and HNO3, as well as the results of modeling F0AM. The article contains figures which reflect the text and important research results, but there are a few moments that require correction:

General recommendation for the entire text of the article: use instead of the word study the word research and its derivatives

Line 10-11: Abstract: Nitrogen oxide (NOX) in the atmosphere causes oxidation reactions with photochemical 10 radicals and volatile organic compounds, causing ozone (O3) accumulation - duplicate words should be replaced.

Line 41-43: It is known to produce HONO mainly through the gas-liquid heterogeneous reaction of NO2 at night, and studies have been conducted to determine the relationship between NO2 and relative humidity [10-12]. The sentence consists of  2 unrelated parts. It is necessary to reformulate or clarify the main idea, because you provide links to the works.

Line 108-109: approximately 23 days and approximately 28 days. You specify the exact measurement period, so the word "approximately" is superfluous here, or give an explanation why it is approximately.

Line 116-118: The water-soluble gases in the gas sample passing through the gas panel diffused and were absorbed through the membrane towards the deionized water in the liquid panel, flowing in the reverse direction of the gas sample. The sentence contains too many repetitions. It is necessary to reformulate.

Line 135: Because =Whereas

Line 137 and 150: Precursors = compounds (or starting material/matter)?

Line 179-189: State the name of the table clearly. Any explanations concerning it should be moved to the appropriate paragraph of the article

Line 186: The difference between the above air pollutants may be due to differences in the seasonal effects of measurements and anthropogenic pollutant sources at the measurement site, such as distance from roads. The sentence is formulated very unsuccessfully and in a confused way. My option: The difference between the mentioned air pollutants may be due to the seasonal variability of anthropogenic load and the difference in the sources of pollutants themselves.

Figures 3 and 4 must be made in color and larger

In Figure 5 there is a gray area and there is no explanation in the text what it is

For Figures 3,4 and 6, develop one color palette and depict everything according to the selected color scale. Black and white drawings in the form presented in the article are unsuccessful.

Line 407 : Fig. 72. Diurnal pattens of Laero+dep and 𝑮𝑨𝑷𝑯𝑵𝑶3 - correct the typo

You didn't write the "discussions" section. You need to add it.

Best regards

Comments on the Quality of English Language

There are sentences and even paragraphs in the article that should be reformulated both from the point of view of semantic load and from the point of view of grammar.

Author Response

(The authors gave the same response as above.)

Reviewer 5 Report

Comments and Suggestions for Authors

The chemical formation of ozone (O3) and secondary aerosol is studied by measurements of HONO, HNO3, and their precursor gas concentrations in the atmosphere using parallel-plate diffusion scrubber-ion chromatography as well as 0-D box model simulations of NOy in the atmosphere together with formation reactions and conversion mechanisms of HONO and HNO3. Further, the conversion mechanism of HNO3 to aerosols is studied, because aerosolization (heterogeneous) reactions affect humans and plants and lowering visibility of the atmosphere. An innovative mixed model for PM2.5 concentration prediction is proposed which includes three modules: feature selection, clustering, and integrated prediction. This model including an Extreme Learning Machine shell boost the overall prediction performance by enhancing feature correlation, refining data irregularity, and improving model prediction ability. A case study with defined evaluation criteria is performed.

General comments

To validate the proposed prediction model's feasibility and accuracy, the study includes a comparative experiment involving six unique predictive models. So, a model is selected which enhances PM2.5 concentration prediction accuracy. It is concluded that future research will explore diverse basic learners further to identifying the optimal basic learner to enhance the robustness and accuracy of the integrated predictive model.

The scientific methods and assumptions are valid and outlined mainly so that substantial conclusions are reached.

The results are sufficient to support the conclusions.

The description of experiments and analyses is complete and precise to allow their reproduction by fellow scientists.

The quality and information of the figures and tables are fine. But the captions should include more information for understanding without details from the manuscript.

Title and abstract are not complete. The objectives are missing in the title and investigation results as well as information about research methods are missing in the abstract-

The overall presentation is well structured and clear.

The mathematical symbols, abbreviations, and units are generally correctly defined and used.

Specific Comments

Chapter Discussion should include all results of this study.

The number of references is relatively low. Can it be enhanced?

Technical corrections

Line 462: Reference is incomplete.

Comments on the Quality of English Language

English language is fine.

Author Response

(The authors gave the same response as above.)

Round 2

Reviewer 2 Report

Comments and Suggestions for Authors

The authors had revised their manuscript carefully, now it is acceptable as it is.

Author Response

Dear Reviewer,

I trust this message finds you well. I wanted to express my sincere gratitude for your thorough and insightful review of my manuscript. Your valuable feedback has been immensely helpful in improving the quality of the paper.

I appreciate the time and effort you dedicated to providing constructive comments and suggestions. Your expertise and attention to detail have significantly contributed to refining the content and addressing the concerns raised during the review process.

I am pleased to inform you that I have carefully incorporated your suggestions into the revised version of the manuscript. Your guidance has been instrumental in enhancing the overall clarity and coherence of the paper.

With your valuable input, I believe the manuscript now meets the necessary requirements for publication. I am grateful for your support and constructive criticism, which have undoubtedly played a crucial role in shaping the final version of the paper.

Once again, thank you for your time, commitment, and expertise. I look forward to the possibility of seeing the improved manuscript published, and I am grateful for your contribution to the advancement of this research.

Best regards,

Authors.

Reviewer 3 Report

Comments and Suggestions for Authors

Minor English revision is requested before potential publication 

Comments on the Quality of English Language

Minor English revision

Author Response

Dear Reviewers,

Thank you very much for your thorough review of our manuscript. We appreciate the valuable feedback and have carefully addressed each comment. In particular, we made minor revisions to enhance the clarity and coherence of the text, and we also paid close attention to refining the English language throughout the paper.

Your insights have been instrumental in improving the overall quality of the manuscript, and we believe the revisions contribute to a more robust and well-structured presentation of our research. We have attached a document outlining the detailed changes made in response to your comments.

We sincerely appreciate the time and effort you dedicated to reviewing our work, and we believe the manuscript is now better positioned for publication. Your constructive feedback has been invaluable, and we are grateful for your thoughtful consideration.

Thank you once again for your commitment to ensuring the quality of scientific literature.

Best regards,

Authors.

Reviewer 5 Report

Comments and Suggestions for Authors

The authors followed the reviewer requirements.

Author Response

(The authors gave the same response as above.)
